# Numerical Simulation of Seed-Movement Characteristics in New Maize Delivery Device

**Rui Liu [1,2], Lijing Liu [1,2,*], Yanjun Li [3], Zhongjun Liu [2], Jinhui Zhao [2], Yunqiang Liu [2] and Xuedong Zhang [2]**

[1] College of Engineering, China Agricultural University, Beijing 100083, China
[2] The State Key Laboratory of Soil, Plant and Machine System Technology, Beijing 100083, China
[3] School of Machinery and Automation, Weifang University, Weifang 261000, China
[*] Correspondence: liulijing@caams.org.cn; Tel.: +86-136-5138-0575

**Abstract:** The delivery device is one of the key components in ensuring uniform grain spacing and achieving high-speed precision seeding. In this paper, a new type of high-speed airflow-assisted delivery device for maize is presented. The gas–solid flow in the delivery device was numerically studied by the coupling method of CFD and DEM. The influence of the structural parameters of the delivery device on the movement of the seeds and the airflow field was analyzed in detail. The matching relationship between the inlet-airflow velocity and the operating speed of the seeder was explored. The results show that the position of the intake seed chamber mainly affects the negative pressure in the distribution area of the mixing chamber. The increase in the shrinkage angle results in the decrease in pressure loss and the decrease in airflow velocity in the delivery chamber. As the diffusion angle increases, the airflow forms a stable straight jet flow and the airflow velocity in the delivery chamber increases. As the ejection angle increases, the bouncing degree of the seed decreases, thereby ensuring the consistency of the seed-ejection direction. The research results show that, when the intake seed chamber is located in the middle, the shrinkage angle is $70°$, the diffusion angle is $30°$, and the exit angle is $60°$, the air-assisted delivery device has better performance. With the increase in inlet wind speed, the seed-ejection speed can also be increased according to a certain proportion, which can meet the requirements of zero-speed seeding and ensure the uniformity of seed spacing, providing a new seed delivery scheme. In the future, if invasive damage to the seed shell is guaranteed to be minimized in high-speed airflow, the new delivery device can meet the requirements of precision seeding under high-speed conditions.

**Keywords:** delivery device; simulation analysis; high speed; airflow field; Zhengdan 958



## 1. Introduction

As one of the three most important grains in China, maize production directly affects our country's food security [1]. When the seeds poured into the seed hopper are all viable improved seeds, precision planting can sow according to the density needed for maize growth, reduce the empty stalk and spikelet rate, and increase maize yield [2]. With the large-scale development of agricultural production, high-speed and high-performance precision planters are required.

The core process of high-speed and high-precision planting technology is as follows. In the first step, the seed-metering device picks up a single seed from the seed hopper and sequentially delivers it to a seed-delivery device. In the second step, the seed-delivery device transports the seeds to the seed trench to distribute them according to a predetermined arrangement [3,4]. Under high-speed working conditions, the seed meter is prone to seed leakage. In order to reduce the leakage rate, researchers have designed the seed metering device with high-speed centrifugal filling-clearing technology, a hill-drop seed-metering device with a combination of positive–negative pressure and hole wheel, a disturbance auxiliary seed metering device, etc., [5–7]. At present, the passive seed-delivery tube, which

is widely used, depends on the curve of the seed-delivery tube to restrict the trajectory, landing speed, and angle of the seeds; the seeds are cast under the action of gravity. When working at high speed, the speed of the seeds increases from the discharge port of the seed-metering device, and the collision between the seed and the seed delivery tube is intensified. Moreover, the combined velocity of the seeds in the horizontal direction is not zero and the anisotropy of the landing velocity and the angle increases, which leads to significant decreases in the seed-spacing-qualification index. In order to solve the problem of the poor seed-spacing uniformity of the seed planter in high-speed operations, Liu et al. [8] designed a motor driving belt-type seed-delivery system, but the position of the dial finger is not adjustable, which limits the accuracy of the operation. Zhao et al. [9] designed a kind of seed guide component which works together with V-shaped groove and flexible seed plucking wheel. When the forward speed of the machine is less than 8 km/h, the uniformity and stability of seed guide are good. Liu et al. [10] took the seed guide tube developed by John Deere Company as the object and used the reverse engineering design software Geomagic Design to realize the localization of foreign mature seed guide tube. When the operation speed is 6.6 km/h, the qualified index of grain distance is 95%. Vaderstad Company of Sweden designed a kind of circular seed guide tube composed of straight-line section and curve section, which uses positive pressure air flow to carry seeds to realize fast and accurate seeding [11]. The Speed Tube conveyor belt seed guide device is produced by Precision Planting Company in USA. The conveyor belt matches the forward speed of the planter, which ensures the uniformity of grain spacing under different working speeds [12]. At present, most of the relevant research focuses on the seed-metering device. By contrast, there are very few studies on the delivery device suitable for high-speed operation, and the operational performance needs to be further improved. Therefore, this paper is based on the low energy consumption of the venturi tube [13–17], which has the unique characteristics of adsorption and the rapid conveying of materials. A new type of high-speed air-assisted delivery device was designed and the working and structural parameters were optimized, which flexibly restrain the trajectory of the seeds.

The discrete element method (DEM) and computational fluid dynamics (CFD) coupling method were successfully applied to study the motion process of particles. Han et al. [18] used the DEM-CFD coupling approach to simulate the working process of the inside-filling air-blowing seed-metering device and optimize the structural parameters. Lei et al. [19] used the DEM-CFD coupling method to reproduce the movement trajectory of the particles, which was used to understand the physical phenomenon of seed movement in an air field. Combined with the DEM-CFD coupling simulation experiment, Wang et al. [20,21] established the motion model of the seeding process and optimized the design of the wheat seeding and seeding machine. In order to study air-seeder dynamic features in detail, Guzman et al. [22] simulated the movement of seeds in an air-seeder distributor system. With the help of DEM-CFD, the seed-feeding device and horizontal seed supply pipe of an air-assisted centralized metering system were designed [23,24]. In order to optimize the performance of a jet-mill, Brosh et al. [25] used the DEM-CFD coupling approach to simulate the fatigue and breakage of particles. When the air-assisted seed passes through the curved pipe, it microscopically reproduces the interaction between particles and solids, and particles and fluids, with the help of DEM-CFD [26]. Kuang [27] established a DEM-CFD model for simulating large-scale conveying systems, which was used to simulate the roping phenomena of the system when the pipeline bending degree was different. The DEM-CFD coupling method also provides support for the design and optimization of corn sowing devices, such as seed disk [28], seed cleaning mechanism [29], unloading mechanism [30], and so on. In this study, DEM-CFD was used to simulate the particle-movement process in the seed-delivery device. The reliability of the simulation was verified by comparing the rig test data with the simulation test data. From the flow field and the movement process of the seeds, the shrinkage angle and diffusion angle of the venturi, the position of the intake seed chamber, and the ejection angle of the seeds were analyzed. In order to ensure the adaptability of the seed-delivery device and the seed-metering device at different working speeds, the effects of different inlet wind speeds

on the seed-ejection speed were studied. In order to meet the demand of the high-speed operation of the planter, the design of a new type of air-assisted conveying device is presented in this paper, which improves the operational precision of the planter.

## 2. The Seed-Guiding System and Working Process

*Overall Structure of Seed-Metering Device*

The structure of the high-speed seed-metering device and the air-assisted delivery device is shown in Figure 1. It is mainly composed of a case, lower seed cleaning knife, upper seed cleaning knife, seed disk, negative pressure chamber, air barrier plate, seed-delivery device, and forced seed cleaning knife. When the sowing system is working, the seed metering device relies on negative pressure to pick up the seeds and carries single seeds with the assistance of the seed-cleaning knife. Arriving at the discharge port, the seeds enter the conveying device under the action of gravity. In the seed-delivery device, the positive air pressure blows the seeds into the seed trench.

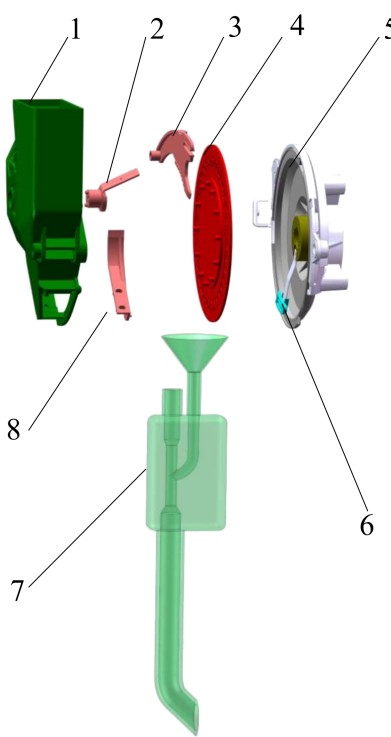

**Figure 1.** Air-assisted high-speed precision seeding and seed-delivery system: (1) case, (2) lower clearing knife, (3) upper seed cleaning knife, (4) seed disk, (5) negative pressure chamber, (6) air barrier plate, (7) seed-delivery device, (8) forced cleaning knife.

As shown in Figure 2a, a new type of conveying device was designed based on the working principle of the Venturi tube structure. The main structural parameters include the diameter of the air inlet chamber ($D_1$), the diameter of the intake seed chamber ($D_2$), the diameter of the mixing chamber ($D_3$), the diameter of the delivery chamber ($D_4$), the shrinkage angle between the air-inlet chamber, and the mixing chamber ($\gamma_1$), the diffusion angle between the mixing chamber and the delivery chamber ($\gamma_2$), the ejection angle ($\gamma_3$), the mixing-chamber length ($L_1$), and the delivery-chamber length ($L_2$). The movement process of the seeds in the air-assisted delivery device can be divided into four sections: negative-pressure adsorption acceleration, section I; pressure-transition deceleration, section II; positive-pressure stable acceleration, section III; and constant-speed seed feeding, section IV. As shown in Figure 2b, the seed colors represent different seed motion speeds. When the pipe diameter changes from large to small, that is, when the shrinkage angle is $\gamma_1$, the velocity of the high-speed airflow increases, the pressure

decreases, and negative pressure is produced. In section I, the airflow adsorbs and entrains the seeds to accelerate their movement. It should be noted that the position of the intake seed chamber affects the adsorption movement of the seeds. When the pipe diameter changes from small to large, that is, when the diffusion angle is $\gamma_2$, the velocity of the high-speed airflow decreases and the pressure increases. Due to the transition of the pressure from negative to positive, under the action of the reverse pressure of section II, the seeds decelerate; the seeds then enter stable-acceleration section III and perform acceleration movement. When the seed speed value in the horizontal direction is close to the forward speed value of the seed planter, the seed speed meets the predetermined value requirement, and the zero-speed seeding is completed in section IV.

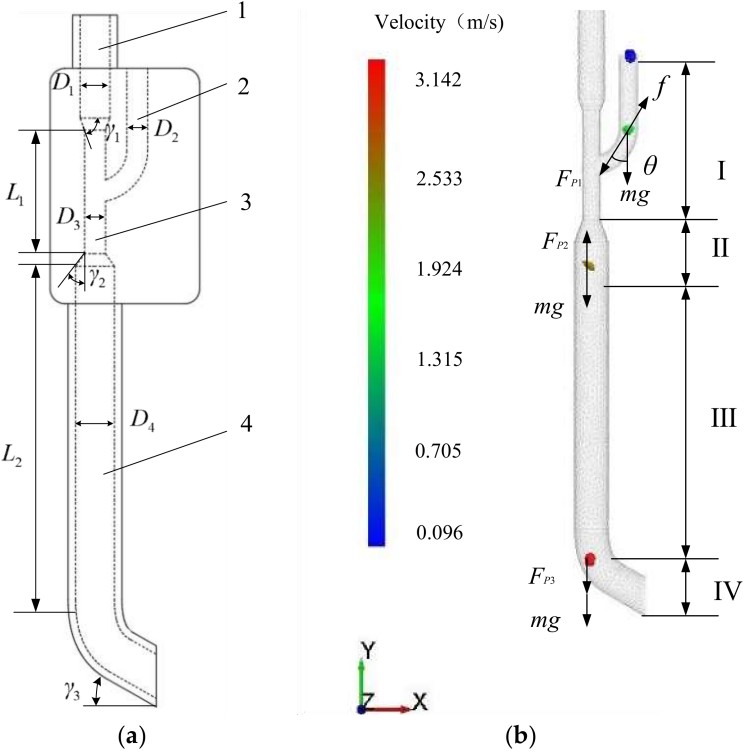

**Figure 2.** Schematic diagram of the structure of the seed-delivery system. (**a**) 1—Air inlet chamber, 2—intake seed chamber, 3—mixing chamber, and 4—delivery chamber. (**b**) Analysis of seed movement process: I adsorption-acceleration section, II deceleration section, III stable-acceleration section, and IV constant-speed seeding section.

According to the momentum theorem, the dynamic equation of the seed in section I can be obtained as

$$mv_1 = \int_0^{t_1} (F_{P1} + mg)d_t - \int_0^t fd_t$$
$$F_{P1} = \tfrac{4}{3}\pi(\tfrac{1}{2}d)^3 \Delta P_1 \tag{1}$$

where $t_1$ is the movement time of the seeds in section I, s; $t$ is the time for the frictional force to work, s; $m$ is the seed mass, kg; g is the acceleration of gravity, m/s$^2$; $f$ is the frictional force of the seed, N; $F_{P1}$ is the pressure-gradient force of the seed in the adsorption section, N; $d$ is the seed diameter, m; $\Delta P_1$ is differential pressure of the seed in the adsorption section, Pa; $v_1$ is the speed of the seeds in a section I, m/s.

The kinetic equation of seeds in stage II is given by

$$mv_2 - mv_1 = \int_{t_1}^{t_2} (mg - F_{P2})d_t$$
$$F_{P2} = \tfrac{4}{3}\pi(\tfrac{1}{2}d)^3 \Delta P_2 \tag{2}$$

where $v_2$ is the speed of the seeds in section II, m/s; $t_2$ is the movement time of the seeds in section II, s; $F_{P1}$ is the pressure gradient force of the seeds in the deceleration section, $N$; $\Delta P_2$ is differential pressure of the seeds in the deceleration section, $P_a$.

When the seeds are stably accelerating in section III, the high-speed airflow exerts a drag force on the seeds [19]. Under the synergistic effect of the drag force and gravity, the dynamic equation of the seeds could be expressed as

$$mv_3 - mv_2 = \int_{t_2}^{t_3} (mg + F_{P3}) d_t$$
$$F_{P3} = \frac{3}{4} C_D m \rho_a \frac{(\vec{v}_a - \vec{v}_p) |\vec{v}_a - \vec{v}_p|}{\rho_p d} \tag{3}$$

$C_D$ is the coefficient of drag force calculated according to Sommerfeld.

$$C_D = \begin{cases} \frac{24}{\mathrm{Re}_p} (\mathrm{Re}_p \leq 1) \\ \frac{24}{\mathrm{Re}_p} (1 + 0.15 \mathrm{Re}_P^{0.678})(1 < \mathrm{Re}_P \leq 1000) \\ 0.44 (1000 < \mathrm{Re}_P \leq 2 \times 10^5) \end{cases} \tag{4}$$

$$\mathrm{Re}_P = \frac{\kappa \rho_a d |v_a - v_p|}{u_g} \tag{5}$$

where $t_3$ is the movement time of the seeds in section III, s; $F_{P1}$ is the pressure gradient force of the seeds in the deceleration section, N; $\rho_a$ is the gas density, kg/m$^3$; $\rho_p$ is the maize density, kg/m$^3$; $v_a$ is the air density, m/s; $v_p$ is the seed density, m/s; $v_3$ is the speed of the seeds in section III, m/s.

The flow time of seeds in section IV is short; the momentum theorem shows that the velocity increase of seeds in this section is small and can be ignored. For this reason, the ejection velocity of the seeds at the end of the delivery device is calculated from the following equation:

$$v = v_3 \tag{6}$$

A comprehensive analysis of the movement process of the seeds shows that the negative pressure at the lower part of the intake seed chamber and the mixing chamber helps the seeds to enter the delivery device quickly. The reverse-pressure difference between the lower part of the mixing chamber and the upper part of the delivery chamber is small and the airflow speed in the delivery chamber is large, which is beneficial for obtaining the ideal ejection speed of the seeds. Taking the ejection velocity of the seeds as the evaluation index, the airflow velocity required for different working speeds under the same structural parameters was analyzed.

### 3. Materials and Methods for Simulation Experiments

#### 3.1. Seed Modeling

Maize seeds can be classified as spheroid-shaped, horse-tooth-shaped and ellipsoid cone shape according to their shapes. Because of the different seed shapes, the multi-ball method was used to establish the maize-seed model in EDEM software, which can more realistically simulate the surface conditions of particles, such as smoothness and roughness [31]. Furthermore, the accuracy of the model is ensured and the efficiency of simulation calculation is improved [32,33]. In this paper, Zhengdan 958 maize seeds were selected as the modeling object, the length range of seeds were 8.91~10.68 mm, the width range were 7.2~8.15 mm, and the thickness range were 5.72~6.42 mm. The seed model was obtained by the multi-sphere method, as shown in Figure 3.

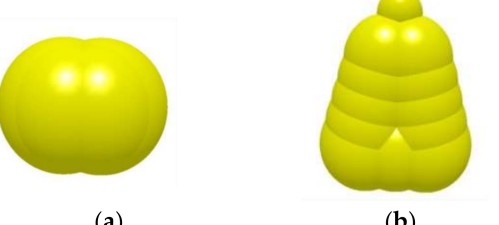
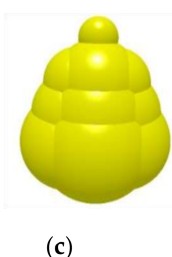

(**a**)                    (**b**)                    (**c**)

**Figure 3.** Simulation seed model. (**a**) Spheroid shape, (**b**) horse-tooth shape, (**c**) ellipsoid cone shape.

*3.2. Numerical Calculation Methods and Boundary Conditions*

In the DEM-CFD coupling simulation, the ICEM module in ANSYS software was used to divide the fluid domain into tetrahedral CFD cells. The total number of meshes was 70,891, and the minimum size of the mesh model was $5 \times 10^{-6}$ m. The inlet of the air flow was set as the velocity inlet, and the outlet of the air flow was set as the pressure outlet. The realizable k-ε turbulence model was used to calculate the fluid motion. In the simulation test, since the collision between the particle and the wall and the rotation of the particle itself needs to be considered, the Euler–Lagrange method was applied to the DEM-CFD coupling interface. Since the particle deformation is not considered in the simulation, Hertz–Mindlin (no slip) was used as the contact model of the particles in EDEM. The two software time-step settings must satisfy the fact that the time step in FLUENT is an integer multiple of the time step in EDEM, so that the two-way transmission and feedback of data can be realized between the software. The time step in EDEM was set to $2 \times 10^{-6}$, the time step in Fluent was set to $1 \times 10^{-4}$, and the total simulation time was 2 s.

*3.3. Simulation Test Design*

In order to determine the appropriate position of the intake seed chamber, the upper, middle, and lower parts of the intake seed chamber were simulated, and the changes in the airflow field and the velocity of the seeds were studied. The shrinkage angle determines the air pressure in the intake seed chamber and the mixing chamber. In order to obtain the best shrinkage angle, the simulated shrinkage angles were 60°, 70°, and 80°. After obtaining the suitable shrinkage angle and the position of the intake seed chamber, the effects of the diffusion angle of four levels: 10°, 20°, 30° and 40°, and the ejection angle of four levels, 20°, 30°, 40°, 50° and 60°, on the seed-exit velocity were studied. In these simulations, the inlet velocity of the airflow was 15 m/s, and the seed-generation rate at the EDEM kernel factory was 9 seeds per second, corresponding to a planter working speed of 8 km/h. In light of the agronomic requirements of maize planting, in order to meet the needs of high-speed precision planting, the matching relationship between the seed-ejection speed and the airflow speed was studied when the working speeds of the seeder were 10 km/h, 12 km/h, 14 km/h, and 16 km/h, respectively. The main parameter values required in the simulation test are shown in Table 1 [34,35].

*3.4. Model Validation*

In order to verify the accuracy of the simulation test, a rig test was carried out. The test device is shown in Figure 4. The working parameters were the same as those of the simulation test. Under different inlet airflow speeds, a high-speed camera was used to record the movement process of the seeds. The video was saved in raw.4 format and imported into the motion-analysis software TEMA Classic. The first step was to select the marker tracking point and then perform algorithm selection and parameter settings for this point. The second step selected the seed-motion section to identify and interpret the tracking point. The third step was to calibrate the seed-motion section in two dimensions. The fourth step was to display the tracking data, obtain the two-dimensional pixel coordinates of the marked maize seeds in the image sequence, and calculate the ejection speed. The air velocity at the outlet of the delivery device was measured by a high-precision anemometer. Figure 5

is a comparison diagram of the simulation-test and bench-test results, the agreement between the simulation value and the measured value of the outlet airflow velocity was 94.7%, the coincidence degree between the simulation value and the measured value of the seed exit velocity was 92.1%. The simulation results were in good agreement with the experimental data, which shows that it is reliable to use the DEM-CFD coupling method to study the delivery device.

**Table 1.** Computational parameters used in the simulation.

| | Parameter | Maize | Organic Glass |
|---|---|---|---|
| Solid phase | Poisson's ratio | 0.4 | 0.5 |
| | Shear modulus/Pa) | $1.37 \times 10^8$ | $1.77 \times 10^8$ |
| | Density/kg·m$^{-3}$ | 1197 | 1180 |
| | Restitution coefficient (with seed) | 0.182 | 0.621 |
| | Static friction coefficient (with seed) | 0.431 | 0.459 |
| | Rolling friction coefficient (with seed) | 0.0782 | 0.0931 |
| Gas phase | Fluid | Air | |
| | Gravitational acceleration/m·s$^{-2}$ | 9.81 | |
| | Density/kg·m$^{-3}$ | 1.225 | |
| | Dynamic viscosity/Pa· s | $1.7894 \times 10^{-5}$ | |

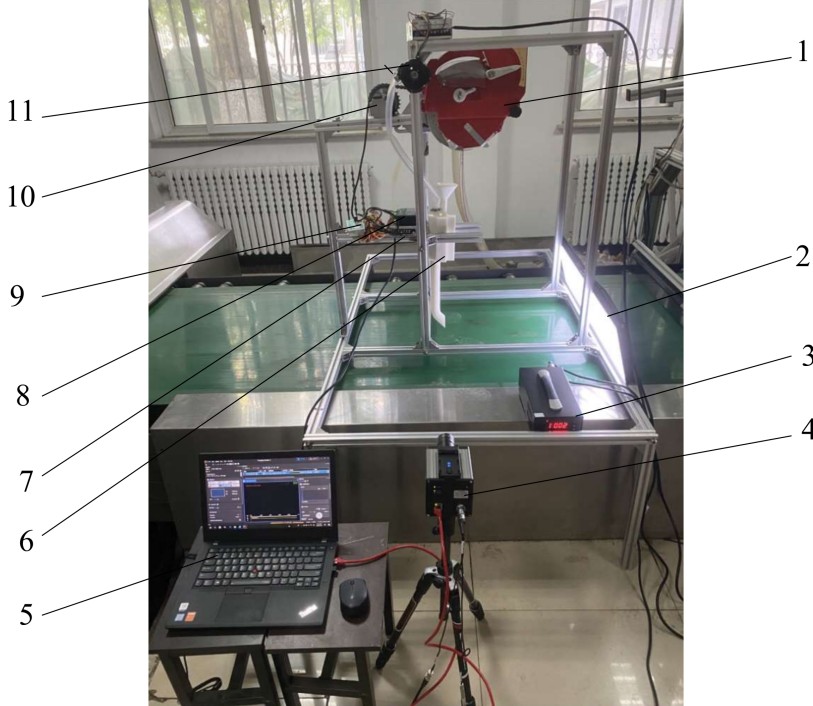

**Figure 4.** Experimental facility of high-speed seed-delivery system. (1) Seed-metering device, (2) fill light, (3) controller of fill light, (4) L-PRI 1000 high-speed camera (produced by AOS Technologies AG, with a frame rate of 500 frames per second during the test), (5) PC, (6) seed-delivery system, (7) power source, (8) drive, (9) controller, (10) stepper motor, (11) fan.

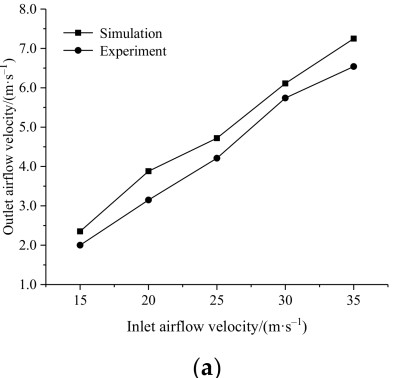
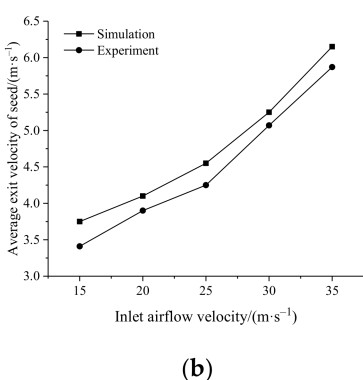

| (**a**) | (**b**) |

**Figure 5.** Comparison between simulation and experimental results. Note: (**a**) relationship between outlet airflow velocity and inlet-airflow velocity; (**b**) relationship between the average exit velocity of seeds and the velocity of inlet air flow.

## 4. Simulation Results and Discussion

Five maize varieties with wide planting areas and large size differences in China were selected. The maximum equivalent diameters of the corn grain were calculated as 10 mm based on their triaxial rulers [36]. The diameter of the seeding chamber needed to be such that the seeds could skid smoothly; therefore, the diameter $D_2$ of the seeding chamber was set as 14 mm. The intake seed chamber was directly connected to the mixing chamber. In order to avoid pressure loss due to sudden changes in pipe diameter, $D_3$ was determined as 14 mm. Since the length of the mixing chamber should ensure the delivery efficiency [37], $L_1$ was set as 85 mm. The air-inlet chamber was directly connected to the outlet of the fan. In order to facilitate the calculation of the wind speed, the diameter of the inlet of the air-inlet chamber was determined to be the same as the diameter of the outlet of the fan; $D_1$ was determined as 20 mm. The diameter $D_4$ of the delivery room was 26 mm and the length $L_2$ was 200 mm.

### 4.1. Optimal Design of Intake Seed-Chamber Location

Since the intake seed chamber is directly connected to the mixing chamber, the position of the intake seed chamber affects the size and distribution range of the pressure in the intake seed chamber and the mixing chamber, which in turn affects the working performance of the seed-delivery device. The upper intake seed chamber is located 27.5 mm above the middle position; the lower intake seed chamber is located 27.5 mm below the central position. The shrinkage angle is set to 80°, the diffusion angle is set to 10° and the ejection angle is set to 20°. Under the same experimental conditions, the pressure in the delivery device, the speed of the movement of the seeds, and the passage time of the seeds were analyzed. The airflow pressure and velocity under different positions of the intake seed chamber are shown in Figures 6 and 7, respectively; the variation law of the gas flow field in the feeding chamber was essentially the same. In the mixing chamber, when the position of the intake seed chamber moved from top to bottom, the minimum negative-pressure distribution area in the mixing chamber decreased, the absolute value of negative pressure decreased, and the airflow velocity decreased. In order to quantify the influence of the position of the intake seed chamber on the seed movement, the velocity change and flow time of the seeds in the delivery device were extracted, as shown in Figures 8 and 9. As shown in Figure 8, when the position of the intake seed chamber was different, the change trend of the speed of the seeds was the same, increasing at first, then decreasing, and finally increasing steadily. It is particularly noteworthy that there was little difference in the velocity values of the seeds of the three shapes. In order to conveniently process the data results, the average ejection velocity of the seeds can be used as an evaluation index. When the intake seed chamber was located in the upper part, the cumulative value of seed velocity in the I stage was the highest, and the speed-reduction value of the seeds in the

III stage was the largest. When the intake seed chamber was located in the lower part, the velocity accumulation of the seeds in the I stage was the lowest. When the intake seed chamber was located in the middle, the speed decrease value of the seeds in the II stage was the smallest, and the speed-increase value of the seeds in the III stage was the largest. After the seeds passed through three stages of motion, their ejection speeds were in the order observed when the feeding chamber was located in the middle, upper, and lower parts. It can be seen from Figure 9 that there was a slight difference in the flow times of seeds of different shapes in the seed-delivery device, and the flow times of the ellipsoid cone-shaped seeds were slightly longer than those of the spheroid shape and horse-tooth shape. When the intake seed chamber was in the lower position, the flow time of the seeds of the three shapes was the longest. The flow time of the seeds in the delivery device was long, which further proves that the acceleration effect on the seed was not good. Therefore, the position of the intake seed room should be arranged in the middle of the mixing chamber.

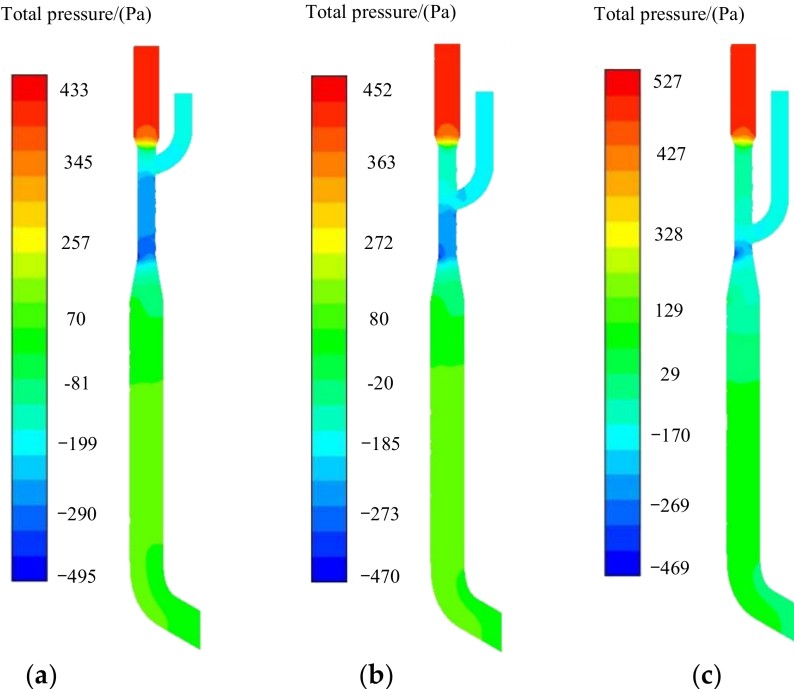

**Figure 6.** Airflow-pressure contours of different intake seed chamber locations. (**a**) Upper, (**b**) central, (**c**) lower.

### 4.2. Effects of Shrinkage Angle

When the intake seed chamber is located in the middle, the diffusion angle is 10° and the exit angle is 20°, Figure 10 shows the airflow pressure and velocity along the *y*-axis in the delivery device under different shrinkage angles. It can be seen that when the shrinkage angle was different, the change trend of the airflow-pressure gradient and velocity along the *y*-axis was the same, the absolute value of the negative pressure of the airflow in the I stage increases gradually and the airflow velocity increased. In the II stage, the airflow pressure increased and the velocity decreased. The pressure and velocity fluctuation of the airflow in stage III and IV were very small. When the shrinkage angle was 60°, the absolute value of negative pressure in section I and the reverse pressure difference in section II were both the largest. With the shrinkage angle was 80°, the absolute value of negative pressure in section I and the airflow velocity in section III were the smallest. In order to further clarify the effect of the shrinkage angle on the seed-ejection velocity, Table 2 shows the average force acting on the seeds and the average ejection velocity of the seeds in each motion stage at different shrinkage angles. When the shrinkage angle was 60°, the seeds received the largest force in each movement stage. With the shrinkage angle was 80°, the

seeds received the smallest force in each movement stage. The average ejection velocity of the seeds was arranged in the order of shrinkage angles of 70°, 60°, and 80°. This may have been due to the fact that the larger the reverse-pressure difference, the greater the resistance to the seeds in the II stage, resulting in a sharp decrease in the speed of the seeds. With the seed entered stage III, the initial velocity was the smallest; therefore, when the shrinkage angle was 60°, the speed of the seeds was lower than when the shrinkage angle was 70°. Therefore, after comprehensive consideration, the shrinkage angle was determined to be 70°.

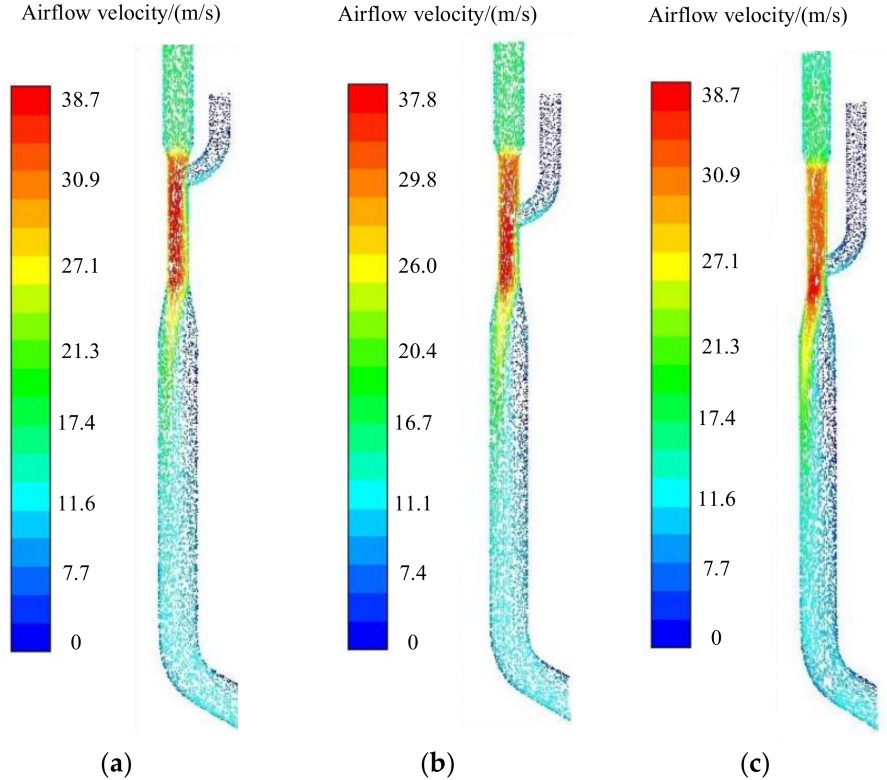

**Figure 7.** Airflow-velocity contours of intake seed chamber locations. (**a**) Upper, (**b**) central, (**c**) lower.

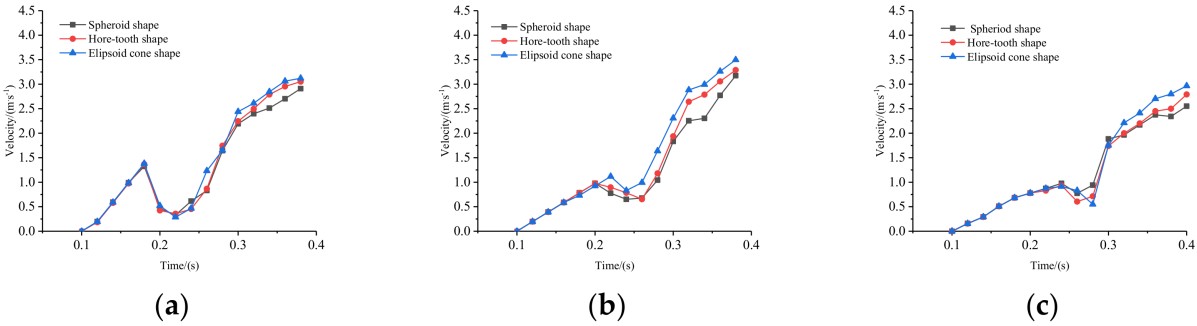

**Figure 8.** Variation of average seed velocities with time under different intake seed chamber positions. (**a**) Upper, (**b**) central, (**c**) lower.

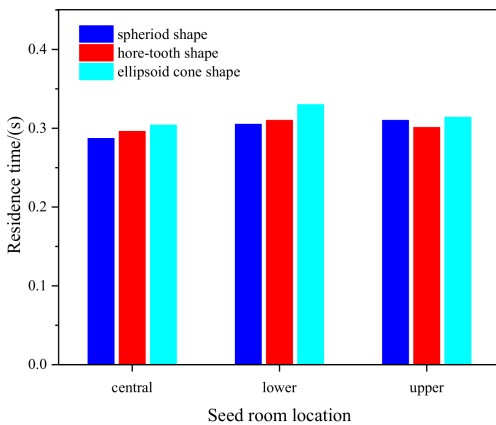

**Figure 9.** Average flow times of seeds in the seed-delivery system.

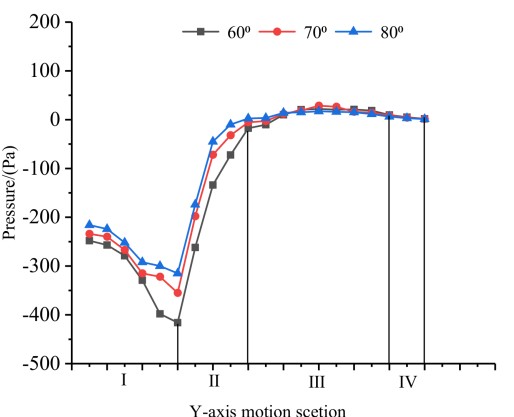
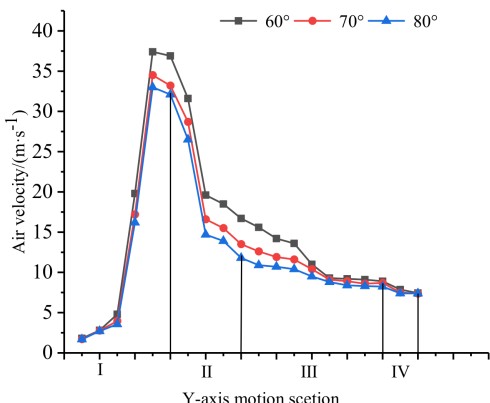

**Figure 10.** Influence of shrinkage angle on air pressure and velocity of conveying device. Note: I adsorption section; II deceleration section; III stable-acceleration section; IV constant-speed seeding section.

**Table 2.** Mean ejection velocities and mean forces at each stage of seed movement.

| Shrinkage Angle (°) | $F_I$ (N) | $F_{Iy}$ (N) | $F_{II}$ (N) | $F_{IIy}$ (N) | $F_{III}$ (N) | $F_{IIIy}$ (N) | $V$ (m·s$^{-1}$) |
|---|---|---|---|---|---|---|---|
| 60 | 0.0052 | 0.0049 | 0.0560 | 0.0536 | 0.0064 | 0.0060 | 3.01 |
| 70 | 0.0045 | 0.0043 | 0.0412 | 0.0409 | 0.0059 | 0.0055 | 3.36 |
| 80 | 0.0035 | 0.0032 | 0.0334 | 0.0332 | 0.0044 | 0.0038 | 2.84 |

### 4.3. Effects of Diffusion Angle

When the intake seed chamber is located in the middle, the shrinkage angle is 70° and the exit angle is 20°, Figures 11 and 12 show the effects of the diffusion angle on the air pressure and velocity in the delivery device, respectively. It can be observed that when the airflow passed through the diffusion angle and entered the delivery chamber from the mixing chamber, the velocity of the airflow decreased and the static pressure increased. With the increase in the diffusion angle, the absolute value of the negative pressure in the lower part of the mixing chamber gradually decreased, the positive pressure in the conveying chamber was roughly the same, and the reverse-pressure gradient between the lower part of the mixing chamber and the upper part of the conveying chamber reduced. When the diffusion angle changed in the range of 10°~20°, the jet line of the gas at the diffusion port was close to the side of the tube wall, with the diffusion angle varying from 30°~40°; the velocity streamline of the gas at the diffusion port was a stable straight jet. In order to further clarify the influence of the diffusion angle on the motion law of the

seeds, the average force acting on the seeds and the average ejection velocity of the seeds in each motion stage were extracted, as shown in Table 3. With the increase in the diffusion angle, the variation trend of the resultant force $F_I$ and the y-direction component force $F_{Iy}$ in sections I and II of the seeds was the same; the resultant force $F_I$ and y-direction component force $F_{Iy}$ of the seeds in the III stage increased at first and then decreased. The ejection velocity of the seeds was the highest when the diffusion angle was $30°$. Based on this analysis, the optimum diffusion angle is $30°$.

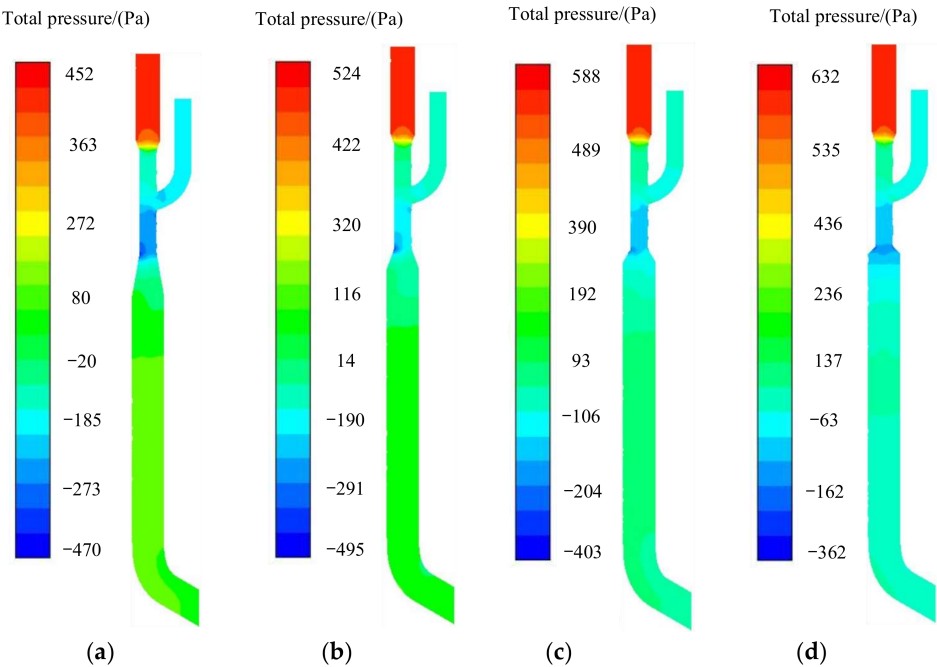

**Figure 11.** Airflow-pressure contours of different diffusion angles. (**a**) $\gamma_2 = 10°$, (**b**) $\gamma_2 = 20°$, (**c**) $\gamma_2 = 30°$, (**d**) $\gamma_2 = 40°$.

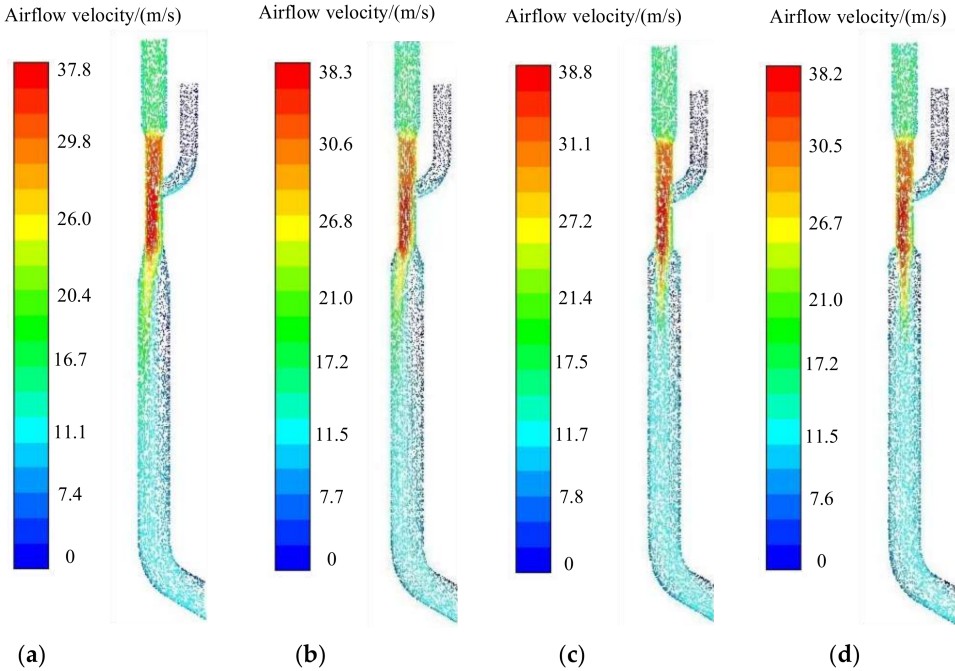

**Figure 12.** Airflow-velocity contours of different diffusion angles. (**a**) $\gamma_2 = 10°$, (**b**) $\gamma_2 = 20°$, (**c**) $\gamma_2 = 30°$, (**d**) $\gamma_2 = 40°$.

**Table 3.** Mean ejection velocities and mean forces at each stage of seed movement.

| Diffusion Angle (°) | $F_I$ (N) | $F_{Iy}$ (N) | $F_{II}$ (N) | $F_{IIy}$ (N) | $F_{III}$ (N) | $F_{IIIy}$ (N) | $v$(m·s$^{-1}$) |
|---|---|---|---|---|---|---|---|
| 10 | 0.0045 | 0.0043 | 0.0462 | 0.0453 | 0.0064 | 0.0061 | 3.21 |
| 20 | 0.0038 | 0.0037 | 0.0289 | 0.0280 | 0.0076 | 0.0073 | 3.48 |
| 30 | 0.0035 | 0.0033 | 0.025 | 0.0249 | 0.0083 | 0.0080 | 3.75 |
| 40 | 0.0029 | 0.0028 | 0.018 | 0.0171 | 0.0072 | 0.0069 | 3.40 |

*4.4. Effects of Ejection Angle*

When the seed moves at the end of the seed delivery device, it collides with the wall and causes bouncing. The heights of the bounces directly affect the ejection directions of the seeds. In order to explore the relationship between the ejection angle and the jumping heights of the seeds, as shown in Figure 13a, when the intake seed chamber is located in the middle, the shrinkage angle is 70°and the diffusion angle is 20°, the vertical bounce distance *h* between the seed and the slope of the tube wall was extracted. Figure 13b shows the bouncing distances of the seeds at different exit angles. With the increase in the ejection angle, the bouncing heights of the seeds decreased. This may have been due to the inclination of the end wall of the seed-delivery device increasing as the ejection angle increased, resulting in a decrease in the force of the seeds hitting the wall. When the ejection angle was 60°, the bouncing distance of the seeds was the shortest, and the difference in the bouncing height of each seed was small, which helped the seeds to enter the seed groove with the same ejection speed and direction, ensuring the uniformity of the grain spacing. Therefore, the ejection angle was determined as 60°.

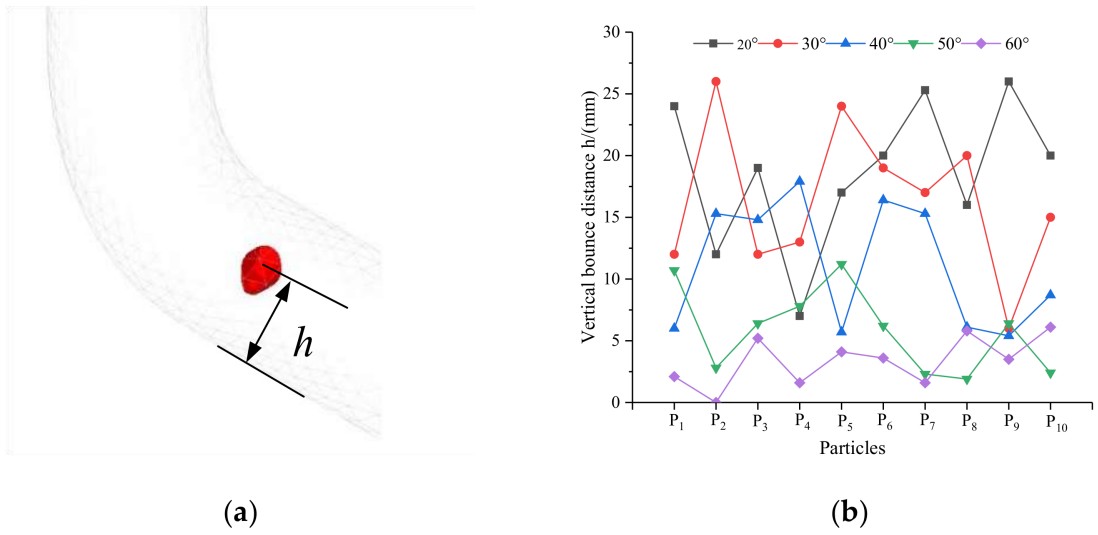

(**a**)  (**b**)

**Figure 13.** Effects of different ejection angles on seed motion: (**a**) defines the bounce heights of the seeds, (**b**) is the bounce heights of seeds with different ejection angles.

*4.5. Effect of Inlet-Airflow Velocity on Seed Motion*

When the working speed of the seed planter is different, different seed-ejection speeds are required to meet the needs of zero-speed seeding. In order to obtain the matching relationship between the working speed of the seed planter and the inlet-airflow velocity, wind speeds of 20, 25, 30, and 35 m/s were selected to study the seed-ejection speed. As shown in Figure 14, with the increase in inlet-airflow velocity, the pressure distribution rules of the intake seed chamber, mixing chamber, air-inlet chamber, and delivery chamber were similar; the positive pressure value of the air-inlet chamber and delivery chamber increased, while the negative pressure of the intake seed chamber and the lower part of the mixing chamber decreased. The reverse-pressure difference between the delivery chamber

and the lower part of the mixing chamber increased. It should be noted that the increase in the reverse-pressure difference directly affected the ejection speeds of the seeds. In order to further analyze the influence of the inlet-airflow velocity on the motions of the seeds, the motion process of the seeds is recorded in Figure 15. As the inlet-airflow velocity increased, the velocity of the seeds increased when the first seed reached the same location. The position of the second seed gradually moved upward and the distance between two adjacent seeds in the *y*-axis direction increased, which further indicated that the greater the inlet-airflow velocity, the greater the movement speeds of the seeds. Therefore, it was proven that the delivery device designed in this paper can overcome the adverse effects caused by the increase in the reverse-pressure difference, which realizes the increase in the air flow to improve the seed-ejection speed. Figure 16 reproduces the trajectory of the seed. Please notice the lines in the black circle in the picture. Each line represents the trajectory of a seed. We can clearly find that at each level of airflow velocity, the bouncing range of the seed is smaller. The trajectory of seed ejection is basically the same, indicating that seed ejection velocity and direction are basically the same.

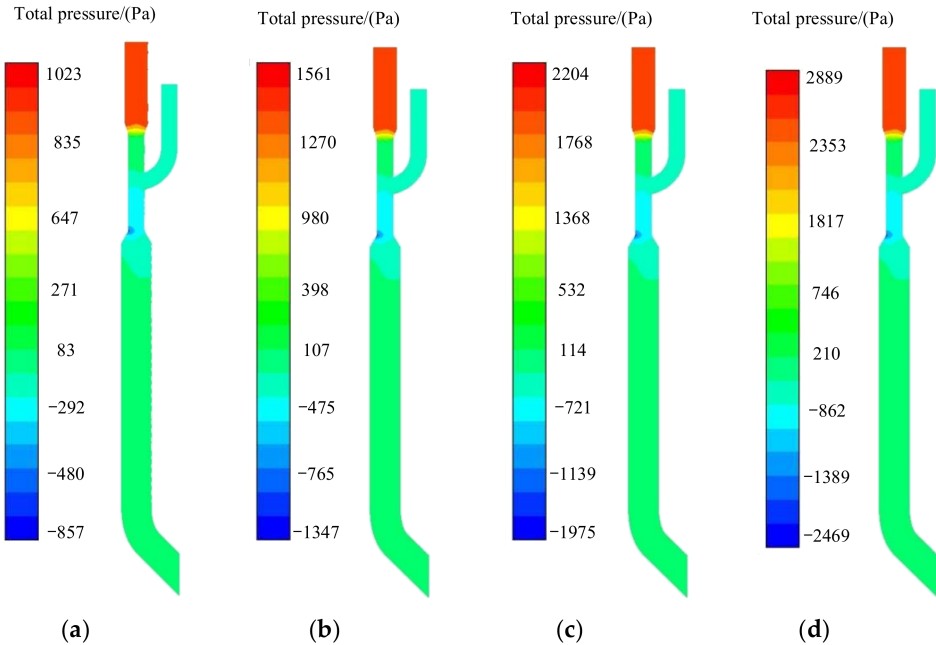

**Figure 14.** Airflow-pressure contours of different inlet-airflow velocities. (**a**) 20 m/s, (**b**) 25 m/s, (**c**) 30 m/s, (**d**) 35 m/s.

Only when the combined speed of the seeds in the horizontal direction is equal to zero they can meet the requirements of zero-speed seeding, as shown in formula (7). According to Equation (7), the average ejection velocity of seeds corresponding to different inlet-airflow velocities is obtained, as shown in Table 4.

$$v_p - v \cos \gamma_3 = 0 \tag{7}$$

where $v_p$ is the forward speed of the seed planter, m/s.

**Table 4.** Average ejection velocity of seeds under different inlet-airflow velocities.

| Inlet-airflow velocity/$m \cdot s^{-1}$ | 15 | 20 | 25 | 30 | 35 |
|---|---|---|---|---|---|
| Average exit velocity of seeds/$m \cdot s^{-1}$ | 3.75 | 4.1 | 4.55 | 5.25 | 6.15 |

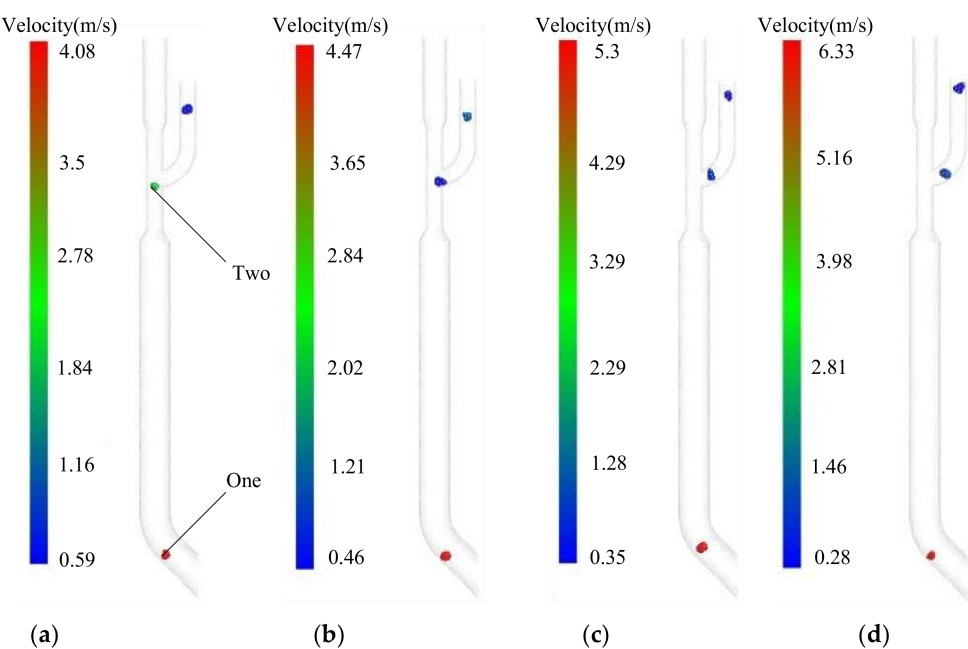

**Figure 15.** Particle motion process in the delivery device under different inlet-airflow velocities. (**a**) 20 m/s, (**b**) 25 m/s, (**c**) 30 m/s, (**d**) 35 m/s.

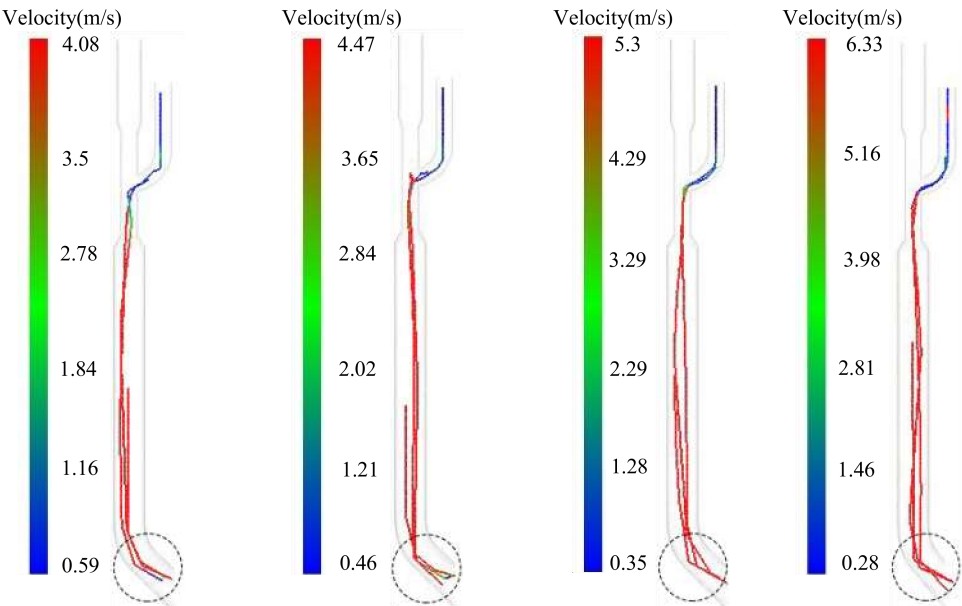

**Figure 16.** Particle motion track in the delivery device under different inlet-airflow velocities.

According to the inlet airflow speed *y* corresponding to the forward speed *x* of the seed planter in Table 4, the matching relationship between them was obtained by using the MATLAB fitting curve, as shown in Equation (8).

$$y = -0.1775x^2 + 8.1x - 54.91 \left(10 \, \text{km·h}^{-1} \leq x \leq 20 \, \text{km·h}^{-1}\right) \tag{8}$$

Note: the fitting-accuracy index of the formula: SSE is 0.8067, $R^2$ is 0.9967, adjusted $R^2$ is 0.9933.

In field operation, limited by the actual working conditions, the forward speed of the seeder cannot always maintain a certain value. When the working speed changes, the speed

of the fan can be adjusted according to Equation (8) to ensure that the inlet airflow speed matches the operating speed. At the same time, for the intelligent seeder, the collaborative operation of the whole machine needs to be controlled online in real time. Equation (8) can also be used to design the control program of working speed, seed meter speed, and fan speed.

### 4.6. Working Performance

According to the requirements of JB/T10293-2013 "Technical conditions of single seed (Precision) seeder", taking the qualified index $\eta$ and the coefficient of variation $C$ of qualified grain distance as the test index, the calculation formulas are respectively as follows.

$$\eta = \frac{n}{N} \times 100\% \tag{9}$$

$$C = \sqrt{\frac{\sum x_i^2}{n_1} - \overline{X}^2} \times 100\% \tag{10}$$

In order to verify the operation performance of the seed-delivery device, the bench test was carried out. The seed feeding height was set as 150 mm, the grain distance was set as 20 cm and 25 cm respectively, and the working speed was 8~16 km/h. The operation performance of the seed-delivery device was studied. The test results are shown in Figure 17.

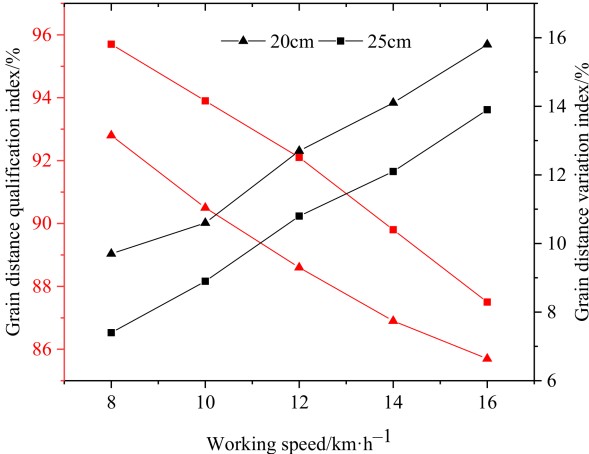

**Figure 17.** Bench test results.

It can be seen from Figure 17 that the working speed is within the range of 8~16 km/h, when the grain distance is 20 cm, the qualification index is not less than 85.7% and the grain distance variation index is not more than 15.8%. When the grain distance is 25 cm, the qualification index is not less than 87.5%, and the grain distance variation index is not more than 13.9%.

In order to further verify the working performance of the positive pressure air-assisted seed-delivery device, a field test was conducted in Beijing Agricultural Machinery Experimental Station of China Agricultural Machinery Academy in 2022. The test power was provided by John Deere 1654 tractor and the test device was a six-row vacuum seeder. The fan of the seed metering device was driven by a hydraulic motor and the fan of the seed guide device was driven by a 24 V battery. The soil type of the experimental site was sandy loam with 13% water content. During the experiment, the theoretical grain spacing was set as 20 cm, the length of the test area was set as 100 m, and the middle 20 m was set as the data acquisition area. The experiment was a single factor experiment, each speed level was repeated three times, and the average value was taken; the test results are shown in Table 5.

**Table 5.** Field test results.

| Working Speed/km·h$^{-1}$ | Qualification Index/% | Variation Index/% |
| --- | --- | --- |
| 9.21 | 89.7 | 11.7 |
| 11.34 | 87.2 | 12.6 |
| 13.07 | 84.5 | 14.7 |
| 15.48 | 81.1 | 17.8 |

Based on the field test results, the airflow-assisted delivery device can meet the requirements of precision seed delivery under high-speed conditions. At each speed level, the qualified index of grain distance was more than 81%, the variation index of grain distance was less than 17.8%. All the test indicators are better than the requirements of the national standard.

## 5. Discussion

From the microscopic point of view, the airflow field, pressure field, seed-movement speed, and seed flow time in the conveying device were analyzed. It was observed that the position of the intake seed chamber directly affected the distribution area of the pressure in the delivery device and the air velocity, which in turn determined the movement speeds and flow times of the seeds in the delivery device. From the dynamic point of view, the influence mechanism of the shrinkage angle and diffusion angle on the seed movement were studied, and the change in seed force in different movement stages could be obtained. By analyzing the bounce heights of the seeds under different ejection angles, the optimal ejection angle was determined. This provided the basis for the design of the delivery device and a new method for the study of the conveying process.

Through the simulation and analysis of the matching relationship between the seed-ejection speed and the inlet-wind speed, the movement process, movement speed, and auxiliary conveying mechanism of the population were observed under various conditions. By comparing the actual operation performance test results with references 8, 9, and 10, it can be found that the working speed has been significantly improved while meeting the qualified index of grain distance. At the same time, it effectively shortens the technological gap with foreign products. This paper provides a new seed-delivery method, which promotes the progress of high-speed precision seeder technology in China. In the future, different materials will be used to make the delivery device study the damage degree of seeds colliding with the delivery device under high-speed airflow, so as to determine the best manufacturing material.

## 6. Conclusions

In this paper, the DEM-CFD coupling method was used to simulate the effects of the position of the intake seed chamber, the diffusion angle, the shrinkage angle, the ejection angle, and the inlet-airflow velocity on the airflow field and seed movement in the delivery device. The DEM-CFD coupling model can describe the motions of seeds in the conveying device effectively. The following conclusions were obtained:

(a) The location of the intake seed chamber had an effect on the distribution area and velocity of the airflow field in the mixing chamber. When the position of the intake seed chamber moves from top to bottom, the minimum negative pressure distribution area in the mixing chamber decreases. Moreover, the negative pressure increases and the airflow velocity decreases. When the intake seed chamber is located in the middle, the seed-ejection speed is the largest; when the intake seed chamber is in the lower position, the flow time of the seeds of the three shapes is the longest.

(b) The effects of the contraction angle and diffusion angle on the airflow field and seed movement were analyzed, as were the effects of the shrinkage angle and diffusion angle on the airflow field and seed movement. When the shrinkage angle increases, the negative pressure in the intake seed chamber and the lower part of the mixing

chamber increases, the reverse pressure difference between the lower part of the mixing chamber and the upper part of the delivery chamber decreases, and the airflow velocity in the delivery chamber decreases. When the shrinkage angle is $70°$, the pressure distribution of the delivery device is better, which helps the seeds to obtain their maximum ejection speed. When the diffusion angle is $30°$, the velocity streamline of the gas at the diffusion port is a stable, straight jet flow; furthermore, the airflow velocity in the delivery chamber is the highest.

(c) The effects of the ejection angle and inlet-wind speed on the seed-exit velocity were analyzed. The results show that when the ejection angle is $60°$, the jumping height of the seeds is the lowest, which helps the seeds to enter the seed groove along the same ejection speed and direction. When the inlet wind speed is increased from 15 m/s to 35 m/s, according to a fixed variable, the ejection speed of the seeds can also be increased, according to a certain proportion. By using the DEM-CFD model, the motion mechanism of the seeds under the action of the air flow can be better understood, and the matching model between the forward speed of the seeder and the inlet velocity of air flow can be obtained. When the operating speed is 10~20 km/h, zero-speed seeding can be realized, and the uniformity of the grain spacing can be improved. In the future, the flexible receiving part between the seed-metering device and the delivery device will be studied further, so that the orderly single seeds from the seed meter can enter the delivery device in an orderly manner. In this way, the operational accuracy of the seeder will be further improved.

**Author Contributions:** Conceptualization, Z.L. and L.L.; methodology, R.L.; software, R.L.; validation, J.Z.; formal analysis, L.R; investigation, Y.L. (Yanjun Li) and R.L.; resources, Y.L. (Yanjun Li) and L.L.; data curation, R.L. and Y.L. (Yanjun Li); writing—original draft preparation, R.L.; writing—review and editing, Y.L. (Yunqiang Liu) and L.L.; visualization, X.Z. and R.L.; supervision, L.L.; project administration, Y.L. (Yunqiang Liu) and L.L.; funding acquisition, Z.L. and L.L. All authors have read and agreed to the published version of the manuscript.

**Funding:** This work was mainly supported by Major Science and Technology Special Task Plan of China Machinery Industry Corporation Limited (ZDZX2020-2).

**Institutional Review Board Statement:** Not applicable.

**Data Availability Statement:** Data are contained within the article. The data presented in this study can be requested from the authors.

**Conflicts of Interest:** The authors declare no conflict of interest.

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
