# Peer review of "Numerical Simulation of Seed-Movement Characteristics in New Maize Delivery Device"

_agriculture, doi:10.3390/agriculture12111944_

Round 1
Reviewer 1 Report
(1) The simulation experiments involved are single-factor tests and lack orthogonal tests in the paper, and it is debatable whether better parameter combinations can be obtained.
(2) The actual seeding trials need to be supplemented. The absence of test result data is not sufficient to prove that the uniformity of seed spacing is improved with the high-speed airflow-assisted delivery device.
(3) The introduction of section lacks literature related to seed delivery devices and requires further supplementation of relevant domestic and international research.
(4) In the â…£ constant-speed seeding section, the seeds appear to bounce due to the collision, which has a significant impact on seed speed and direction, and cannot be ignored. There is currently not enough content and evidence in the paper to show that zero speed seeding can be achieved, and further details should be provided on how zero speed seeding can be achieved.
(5) Depending on the type of seed, the parameters such as seed profile size are supplemented.
(6) The experiments lacked descriptions of the relevant experimental conditions, such as the level at which other factors need to be described when conducting studies on the shrinkage angle.
(7) The analysis is not deep enough in the discussion section,and it is recommended to add content according to the experimental results and relevant domestic and international studies.
(8) There are many errors and contradictions in the paper, it is recommended to sort out the whole paper.
Reviewer 2 Report
As the authors point out, "Precision planting can sow according to the density needed for maize growth, reduce the empty stalk and spikelet rate, and increase maize yield".
1) It should be added here that "reduce the empty stalk and spikelet rate" (see L35) will occur only if viable improved seeds are poured into the seed hopper. Since the single seed feeder feeds all the seeds from the hopper in a row, without determining which of them are empty and which are full-fledged.
With the technological effect of the elements of the feeding device on corn seeds during their sowing, clogging, oiling or dusting is possible after a certain time. This negatively affects the accuracy and performance of the device.
2) How high is the probability of clogging, greasing or dusting of working organs?
When developing a new high-speed device for feeding corn seeds using an air stream, two questions must be answered equally:
3) How much does the QUALITY (viability) of corn seeds change with high-speed air flow, different ejection angles? To do this, as a rule, either a viability test is carried out, or studies using a microscope to assess the area and depth of invasive damage to the seed shell.
4) How much does the NUMBER of seeds passing through the pipeline section of a pneumatic seed drill increase per unit of time?
An adequate answer was received to question 4 in the submitted manuscript, justified using the theory of DEM-CFD modeling.
However, when working with biological objects, which are undoubtedly corn seeds, it is important to maintain a balance between QUALITY and QUANTITY, giving priority to quality.
For a more complete demonstration of all the advantages of the experiment, it is desirable to supplement the Introduction section with information about 5) the practical application of the results of DEM-CFD modeling for sowing corn seeds, as well as to increase reader interest 6) at the end of the Discussion section, disclose future research.
7) It is appropriate to add two sentences at the end of the abstract that answer the questions: "What do the results mean in practice? and what remains unresolved?"
8) It is desirable to include in the keywords the breed (Latin name) of the plant whose seeds were used for field tests.
9) It is desirable to specify DOI in the references.
10) As for the ethical aspects related to seeds, a separate section of the type is usually indicated Data Availability Statement: for example, “[latin name of the seed] seeds were used in this study. The seeds were collected in the following provenances (coordinate) in the fall of Year”.
I believe that the manuscript can be considered for publication in the journal with minor changes.
Reviewer 3 Report
The conveying device is one of the key components to ensure uniform grain spacing and realize high-speed precision sowing. In order to meet the demand of corn high-speed precision sowing, a new type of corn high-speed airflow auxiliary conveying device is designed to solve the problems of seed transportation stability and sowing uniformity under the operating conditions of high-speed corn sowing. The gas-solid flow in the conveying device is numerically studied by using the CFD-DEM coupling method. The influence of the structural parameters of the conveying device on the seed movement and air flow field is analyzed in detail, and the matching relationship between the inlet air velocity and the working speed of the seeder is explored. On the whole, the structure of the paper is complete and clear, the topic selection is of great practical significance, and the designed device is innovative. The research results of this paper preliminarily reveal the movement law and influencing factors of grain transportation in the process of corn sowing. It provides a certain reference for the in-depth study of zero-speed planting.
At the same time, the paper also puts forward the following questions and suggestions:
1. Keywords should not be included in the title.
2. In the current research situation at home and abroad, the analysis of seed introduction mechanism is not comprehensive enough, so it is suggested to increase the current situation analysis of seed introduction mechanism.
3. The simulation results are in good agreement with the experimental data, which shows that it is reliable to use DEM-CFD coupling method to study the conveying device. The clear data should be given here. What is the degree of coincidence?
4. According to the air inlet speed y corresponding to the forward speed x of the seeder in Table 4, the matching relationship between them is obtained by using MATLAB fitting curve. What is the purpose of the relationship? the lack of further analysis of the relationship in this paper, it is suggested to add.
5. The references are not enough, so it is suggested that the relevant references should be supplemented in the research background and experiments.
Round 2
Reviewer 1 Report
Accept in present form